# Preparation and Application of Foaming Agent Based on the Compound System of Short-Chain Fluorocarbon and Soybean Residue Protein

**DOI:** 10.3390/ma15207384

**Published:** 2022-10-21

**Authors:** Ning Song, Zhihe Li, Shaoqing Wang, Yuanliang Xiong

**Affiliations:** 1Agricultural Engineering, School of Agricultural Engineering and Food Science, Shandong Research Center of Engineering and Technology for Clean Energy, Shandong University of Technology, Zibo 255000, China; 2Structural Engineering, School of Civil Engineering, Yantai University, Yantai 264000, China

**Keywords:** soybean residue protein, response surface methodology, FS-50, foam concrete

## Abstract

This study provides a new idea for the design of an advanced foaming agent with soybean residue protein (SRP) as a potential protein source. In order to achieve the most effective foaming performance, we employed the novel approach of response surface methodology (RSM) to improve important process parameters in a hot-alkali experiment. The experimental results showed that the optimum reaction parameters of pH and temperature were pH 10.2 and 50.5 °C, respectively, which, when continued for 3 h, led to the highest foaming property of the SRP foaming agent (486 mL). Based on the scheme, we also designed an experiment whereby we incorporated 1.0g/L FS-50 into the SRP foaming agent (SRP-50) to achieve higher foaming capacity compared with the commercial foaming agent. This foaming agent was cheaper than commercial vegetable protein foaming agents (12 USD/L) at 0.258 USD/L. Meanwhile, the properties of foam concrete prepared using SRP-50 were studied in comparison with a commercial vegetable protein foaming agent (PS). The results demonstrated that the foam prepared using SRP-50 had better stability, and the displacement of the foam decreased by 10% after 10 min. During the curing period, the foam concrete possesseda compressive strength of 5.72 MPa after 28 days, which was an increase from 2.95 MPa before. The aperture of the foam ranged from 100 to 500 μm with the percentage increasing up to 71.5%, which indicated narrower pore-size distribution and finer pore size. In addition, the shrinkage of the foam concrete was also improved. These findings not only achieve the utilization of waste but also provide a new source for protein foaming agents.

## 1. Introduction

Foam concrete is composed of foam solution and different cement mixtures, arranged mechanically in even, poured, and molded shapes. A block of good foam concrete has the characteristics of higher strength, smaller mass, better heat protection, and stronger insulation capability [1,2,3,4]. There is a strong correlation between the type of foaming agent used and the quality of the foam concrete [5], which means using a suitable foaming agent is one of the most important factors in the preparation of high-quality foam concrete. Currently, synthetic foaming agents and protein foaming agents have dominated the market [4] as they can make a stable foam by reducing the surface tension of the foaming liquid. Stable foam requires a high concentration of synthetic agents and of the amounts of foaming agents used [6]. Although the foaming ability of the protein foaming agent is lower than that of the synthetic foaming agent, it is much more stable in the performance that researchers are looking for [7]. In addition, protein foaming agents are also used for foam extinguishing agents due to their safety and environmentally friendly nature. As such, there has been a great deal of attention given to the development and improvement of concrete foaming agents, consisting of surfactants and proteins as the main components of the foaming agent [4,7,8,9].

The preparation of foaming agents has so far been studied using a variety of animal and plant proteins [10,11,12,13]. These agents, however, are not sustainable in the long term. Therefore, alternative sources of protein are needed that are environmentally friendly, such as raw materials extracted from sludge [14], bacterial cells [15], and distiller’s grains [16]. Recycling proteins from waste and raw sources will produce some social benefits. However, the experimental process of extracting protein from them is very cumbersome. For example, when extracting some animal proteins, the procedures require acid or alkali hydrolysis, water purification, pressure filtration, vacuum concentration, air drying, grinding, etc. [17]. The processes of purification and extraction are very difficult to carry out, so more and more scholars have devoted themselves to looking for new protein sources and simple preparation processes, which can be widely commercialized.

At present, China is the largest soybean importer and consumer in the world. In 2020, China’s soybean import volume exceeded 100 million tons. During the processing of bean products, a large amount of residue, bean dregs, is produced. During the processing and production of 1.2 tons of tofu, 1 ton of wet bean dregs is left [18,19]. Because soybean protein contains about 10% insoluble protein, it is difficult to separate in the processing process and will remain in the soybean residue [20]. Studies have shown that the residual protein content in soybean residue is 15.2–33.4% [21]. Fresh soybean dregs contain protein, carbohydrates, and other nutrients, which can provide sufficient nutrition and humidity for microorganisms. Therefore, it can be easily decomposed by microbes, hard to preserve, and used as feed. Treating soybean residue as waste residue will not only pollute the environment, but also waste some soybean protein resources [22].

This study proposes a new method to obtain protein only viathermal alkali treatment of soybean residue, without laborious and cumbersome extraction or purification processes [23,24,25]. At the same time, it has the advantages of reusability and environmental sustainability. In order to optimize reaction conditions such as temperature, pH, and the time needed for thermal alkali treatment, we adopted the response surface methodology (RSM) for the desired purpose. An additive suitable for soybean residue protein (SRP) was found, in this experiment, to enhance the foaming performance of the soybean residue protein (SRP) foaming agent. However, previous research only stayed at the level of preparing protein foaming agents. Thus, this study investigates not only foam but also foam concrete to further optimize the products of SRP foaming agents.

## 2. Materials and Methods

### 2.1. Materials

Fresh bean dregs, sodium hydroxide (analytical grade), and Sinopharm were purchased from the Chemical Reagent Company (Zibo, China). FS-50 and FS-3100, two of the most widely used fluorocarbon short-chain surfactants produced by DuPont, were selected for the purpose of producing high foam. They are amphoteric fluorocarbons and their perfluorinated groups contain only 6 carbon atoms and will not decompose to produce perfluorooctane sulfonate (PFOS), which meets the requirements of the US Environmental Protection Agency (EPA) on the voluntary environmental protection plan for PFOS.

The cement used was Portland cement P.II 52.5, which has a density of 3.65g/cm^3^, a specific surface area of 355 m^2^/kg, and a specific strength of 62 MPa. The compositions of P.II 52.5 are enumerated in Table 1. The plant protein-based surfactants (PS) were obtained from Zhicheng New Building Materials Inc (Nanjing, China). at the price of 12 USD/L. This foaming agent was 4% in concentration.

### 2.2. Experimental Methods

#### 2.2.1. Preparation of SRP Foaming Agent

The fresh wet bean dregs were steam-sterilized for 30 min, dried at 50 °C for 24 h, cooled to 25 °C, and placed in a cool place on standby.

We weighed 20.00 g of bean dregs using an electronic balance and added 200 mL of water.

For the hot alkali experiments, sodium hydroxide (1 mol/L) was added for pH regulation. An electronic constant-temperature water bath was used to heat the beakers to the target reaction temperatures. After a certain time, the residue was separated viastanding, cooling, and suction filtration, and the obtained supernatant was the foaming base liquid. After the solution was prepared, it could be tested for its properties.

#### 2.2.2. Single-factor Test

For the foaming ability, a single-factor test was carried out to check the effect of pH value (9, 10, 11), temperature value (40, 50, 60 °C), and time of hydrolysis (2, 3, 4 h). During the single-factor test, we checked in the order: pH value, temperature value and hydrolysis time. That is to say, when the temperature value and hydrolysis time were concerned, the pH value was fixed.

The effect of pH on foaming ability: To check the effect of an alkaline pH value (9, 10, 11), a hydrolysis temperature of 50 °C and hydrolysis time of 3 h were used to measure the foaming ability.

The effect of hydrolysis temperature on foaming ability: To check the effect of temperature for foaming ability, the conditions were set to a pH value of 10.0, hydrolysis time of 3 h, and hydrolysis temperature values of 40, 50, and 60 °C.

In order to determine whether the effect of hydrolysis time on foaming ability is significant, we set the conditions for the single-factor test at a pH value of 10.0, hydrolysis temperature of 50 °C, and hydrolysis times of 2, 3 and 4 h.

#### 2.2.3. Response Surface Test

Based on the response surface method (RSM), three experimental variables (pH, reaction temperature, and reaction time) were optimized to maximize the foaming ability of the soybean residue protein foaming agent [26].

The range and selection level of the independent variables affecting this experiment are listed in Table 2. The Box–Behnken design (BBD) (V12.0, Stat Ease, MN, USA) was used to design 20 experiments with different values of three variables. A STAT ease program (V11.1.0.1, Stat Ease, MN, USA) was used for the analysis of all experimental data.

#### 2.2.4. Analysis Methods

First, we put 100 mL of the bean dreg protein foaming agent solution into a 1000 mL beaker and heated it to 40 °C. After stirring for five minutes at 1200 rpm, the solution was transferred to a measuring cylinder for analysis [27]. We stipulated that after 5 min of foaming, the volume of soybean residue protein (SRP) foaming agent and the volume of foam were labeled with their foaming properties (mL) [28]. The pH tester (PHS-3E) (Shanghai Shengke Instrument & Equipment Co., Ltd., Shanghai, China) was used in this test. In order to ensure that the deviation of the data did not exceed 5%, three tests were carried out in each experiment.

### 2.3. Preparation of Foam and Foam Concrete

The self-made (SRP) and commercial vegetable protein foaming agents (PS) were mixed with water in their respective containers. An air pressure of 0.4–0.6 MPa was applied to the foaming agent after standing in the foaming machine [29]. In order to determine the density of the fresh foam, a 1 L standard container was immediately filled with the fresh foam and weighed.

In order to achieve optimal performance of foam concrete, we calculated the mix proportion in accordance with its targeted density, since it can be adjusted depending on the designed density and the performance of the foam. At this point, we used the following method to determine the proportions of foam concrete (of 1 m^3^) [4]:(1)$$\rho_{d}=S_{a}m_{c}$$
(2)$$V_{2}=K(V-V_{1})=K[V-(\frac{m_{c}}{\rho_{c}}+\frac{m_{w}}{\rho_{w}})]$$
*ρ_d_* (kg/m^3^) is the target density of the foam concrete. *S_a_* is the empirical coefficient, which is 1.2 for standard 52.5 Portland cement. *m_c_* (kg) and *m_w_* (kg) are the masses of cement and water separately. *V* (m^3^) is the volume of foam concrete, equal to 1 m^3^. *V*_1_ (m^3^) and *V*_2_ (m^3^) are the volumes of cement paste and foam, respectively. *ρ_c_* (kg/m^3^) and *ρ_w_* (kg/m^3^) are the densities of cement and water. In this paper, *m_w_* = 0.5 *m_c_*. *K* is a coefficient, decided by the foam quality.

As shown in Table 3, the mix design of foam concrete can be summarized as follows. FC-PS and FC-SRP-50 correspond to the foam concretes. A stirring speed of 100 revolutions per minute was applied initially to the cement slurry, and this lasted for 30 s. After that, the prepared foam was added to the cement slurry along with the remaining ingredients. Afterwards, the foam was introduced into the cement matrix and stirred at 60–120 revolutions per minute for 180 s. In order for the foam concrete to perform properly, it must not be mixed for an excessive amount of time in order to prevent defoaming of the foam, altered pore structure, and macro-performance problems [1]. At the end, a mold type measuring 100 mm × 100 mm × 100 mm was filled. A standard curing box was used to cure the samples for 28 days after they were removed from the mold (after 24 h). The samples were dried for 24 h at 60 °C after 28 days of standard curing, and their density was determined following this drying process. The procedure followed in the experimental study was the Chinese standard “Foam Concrete” (JG/T 266–2011).

### 2.4. Characteristics of Foam

#### 2.4.1. Viscosity Test

This foam was measured using a rotary viscometer (NDJ-1, Shanghai Changji Geological Instrument Co., Ltd., Shanghai, China) of which the No. 1 rotor was used for measuring the foam’s viscosity; this was carried out in a 500 mL beaker.

#### 2.4.2. Surface Tension Test

It was determined that the surface tension was 25 °C using KRUSS k100 surface tension meter (CRUS Scientific Instruments (Shanghai) Co., Ltd., Shanghai, China) and the Du Nouy ring method was used to make the measurement.

#### 2.4.3. Foam Stability

It was determined that the foam was stable by measuring the weight loss of the foam 5, 10, 30, 60, 120, 180, 240, 360, 720, and 1440 min after the foam had been filled into a 1 L container at 25 °C [30].

#### 2.4.4. Optical Microscopy

The thickness of the foam wall (OM) was determined using an optical microscope (DM 750, Suzhou Ouster Optical Instrument Co., Ltd., Suzhou, China) in order to determine the thickness of the wall.

### 2.5. Properties of Foam Concrete

#### 2.5.1. Compressive Strength

Initially, the samples of foam concrete were baked at 60 °C until the quality of the foam concrete did not change. In accordance with GB/T 11969-2008, a constant loading speed of 1 kN/s was used to determine the compressive strength of the samples. The testing machine model was WDW-300.

#### 2.5.2. Drying Shrinkage

In accordance with GB/T 11969-2008, shrinkage tests were conducted on the foam concrete samples in order to determine their shrinkage results. During this experiment, 40 mm × 40 mm × 160 mm samples were used. After being placed in the standard curing box at their initial lengths, the samples were cured for 3, 7, 14, 21, 28, 60, and 90 days, after which the change in length was measured over time.

#### 2.5.3. Microstructure and Pore Structure

The surface morphological properties of foam concrete were examined using a Quanta 250 scanning electron microscope (FEI, Costa Mesa, CA, USA). There was a 35° angle of output from the analyzer. The range of elements analyzed was Be4–Pu94. A 20 kV accelerating voltage was also available with EDS. In addition, the pore structure of the foam concrete was examined using X-ray computed tomography.

## 3. Results and Discussion

### 3.1. Analysis of Single-Factor Test Results

#### 3.1.1. Effect of PH Value on Foaming Ability

When soybean residue is subjected to increased pH levels, its foaming capacity increases first, and then, decreases after reaching pH 10. As a result, hydrolyzed soybean residue protein is best used at pH level 10 in order to achieve optimal results (Figure 1).

#### 3.1.2. Effect of Temperature on Foaming Hydrolysis Ability

Hydrolysis of the foam became more efficient with the increase in hydrolysis temperature, resulting in a greater degree of foaming hydrolysis. (Figure 2) When the hydrolysis temperature is 50 °C, the degree of hydrolysis of the soybean residue reaches the maximum. When the hydrolysis temperature exceeds 50 °C, the degree of hydrolysis of the soybean residue protein decreases with the increase in hydrolys is temperature, reaching 90.75%. It can be seen that at emperature that is too high or too low will affect the hydrolysis degree of the soybean residue protein. Studies have shown that with a higher hydrolysis temperature, the Maillard reaction has an adverse effect on protein extraction [31]. Therefore, when the hydrolysis temperature is 50 °C, the soybean residue protein reaches a better degree of hydrolysis.

#### 3.1.3. Effect of Hydrolysis Time on Foaming Ability

There is a continuous increase in the foaming capacity of soybean residue protein when a hydrolysis time of less than three hours is used (Figure 3). However, when the hydrolysis time is more than 3 h, the foaming capacity increases slowly. Considering the actual production efficiency, the hydrolysis time of the soybean residue protein is 3 h.

### 3.2. Optimization of Experimental Conditions

#### 3.2.1. Property Fitting and Data Analysis

Evaluation of the foaming agents should be considered in two ways: foaming and foam stability [32]. Previous studies have shown that the stability of foaming agents using protein foaming agents is very high. However, their foaming property is inadequate. A foam’s stability is determined by two factors: the stiffness of the foam film and the time taken for the liquid to evaporate. A foam film prepared by a protein foaming agent has high rigidity and is hard to break, because an active material with high molecular weight has a strong interaction force [12]. Therefore, after analysis, we agreed to take foaming as the measurement standard in this study.

For the comprehensive single-factor test results, the experimental data for the response surface analysis were designed using the BBD design, as shown in Table 4. Different experimental conditions correspond to the foam’s foaming range of 420–489.5 mL. The 20 groups of experimental data that were designed were analyzed using Design Expert software. The empirical relationship between foam stability and the three experimental variables was analyzed using response surface methodology. The fitting model for predicting the foaming property is shown in Formula (3). The independent variables X_1_, X_2_, and X_3_ represent pH, reaction temperature, and reaction time, respectively, and the dependent variable Y represents the bubble foaming property.
Y = 485.22 − 0.2439X_1_ + 0.4916X_2_ + 0.1473X_3_ − 0.8988X_1_X_2_ − 3.59X_1_X_3_ − 0.61X_2_X_3_ − 17.31X_1_^2^−16.42X_2_^2^ − 18.62X_3_^2^(3)

In order to analyze the variance of the data, the RSM was used, and the results can be seen in Table 5. The coefficient of determination (R^2^) indicates the proportion of variance in the data that the model can explain or account for. With an R^2^ value of 0.9847, it is indicated that only 1.53% of the total variation cannot be explained by the model, suggesting reasonable agreement between the observed values and the predicted values. F-values that are greater than 0.05 indicate that the model does not fit, and the value of the variation coefficient (CV) is 0.93%. This is enough to confirm the high reliability of the model. Therefore, if we change the experimental variables, we can use the model to reasonably predict the foaming properties of the bubbles.

Each independent variable’s p-value indicates its significance, with smaller p-values indicating greater significance [33]. The F-value indicates that hydrolysis temperature (X_2_) and pH (X_1_) have the highest influence on protein foaming ability, followed by hydrolysis time (X_3_).

#### 3.2.2. Response Surface Experimental Analysis

The relationship between independent variables and dependent variables is clearly described by fitting the experimental data (Figure 4).

The effects of the independent variables X_1_ and X_2_ on the foaming property areshown in Figure 4a. As shown in the Figure 4, when the value of X_1_ of the independent variable is between 9 and 10, with an increase in ionic concentration, the conditions for foam stability are created. At the same time, the foaming property increased with an increase in the independent variables X_1_ and X_2_. Because protein molecules are hydrophobic, foam liquid membranes can adsorb them, and the foam liquid film can be negatively or positively charged at the same time. Due to the increase in the independent variable X_1_, the liquid film surface can carry the same charge. It is possible to prevent the physical drainage of the foam liquid film via the electrostatic repulsion generated by the impact of the charge on the surface of the liquid film, thereby prolonging the time for foam stabilization [34]. By observing the contour map, it can be concluded that when the independent variable X_2_ is in the range of 45–55 °C and X_1_ is about 10, the performance of the dependent variable is the highest (>480 mL). When the independent variable exceeds this optimal range, the solution environment becomes bad, resulting in a decrease in protein content. Because of the possible Maillard reaction, the foaming capacity of the foam decreases [35]. Additionally, since the *p*-value of 0.5566 shows that the independent variables X_1_ and X_2_ have slight interdependence, the fact that the independent variables X_1_ and X_2_ may interact in a minor manner may not be significant in influencing the dependent variable (Table 5) [24].

Figure 4b depicts the influence of the independent variables X_1_ and X_3_ on the dependent variables. According to the contour map, the dependent variable is the highest (>480 mL) when the independent variable X_1_ is 10 and the independent variable X_3_ is between 2.5 h and 3.5 h. When the independent variable X_1_ is high, the protein will be denatured and will not dissolve in the solution. With an increase in the independent variable X_3_, the value of the dependent variable is always low [36]. Although strong alkali will saponify the membrane lipid, the effect on the dependent variable is not obvious. The dependent variable is improved only when the independent variable X_3_ is about 3 h. However, when the independent variable X_1_ is low, it is conducive to the breaking of cell wall with an increase in reaction time; however, when the reaction time is too long, it may lead to the denaturation of protein molecules, so it is not conducive to the improvement of the dependent variable.

The interaction between the independent variables X_2_ and X_3_ is obvious (Figure 4). When the reaction temperature is in a relatively soft environment, the independent variable X_3_changes between 2.5 h and 3.5 h, and the value of the dependent variable is the highest. When the reaction solution is hot and alkaline, it will not only reduce the high-temperature resistance of the cell wall, but will also accelerate the hydrolysis of organic matter, resulting in the rapid rupture of the cell wall [14]. The higher the degree of cell wall rupture, the more protein molecules release, so a large number of protein molecules will produce more bubbles. It should be noted that the higher independent variables, X_1_ and X_3_, will lead to protein molecule failure. With the bond breaking within and between protein molecules, a large number of hydrophobic amino acids will appear in the solution. This reaction will improve the softness and hydrophobicity of the protein molecules [37]. However, when the independent variable X_2_ is greater than 55 °C, no matter what the reaction temperature is, the dependent variable will decrease significantly. When the reaction environment becomes bad, it will not only reduce the value of the dependent variable, but will also produce an unpleasant smell of ammonia because the protein is over-hydrolyzed.

As mentioned above, the changes in X_1_, X_2_, and X_3_ can significantly promote cell fragmentation and protein degradation, so as to improve the foaming property of the product. It should be noted that if the variables involved in the reaction exceed a certain value, some adverse reactions may occur and the dependent variable will be reduced.

#### 3.2.3. Model Verification and Adjustment

According to the results of the response surface analysis, when the independent variable X_1_ of the model is 10.197, X_2_ is 50.538 °C, and X_3_ is 3.105 h, the highest dependent variable (491.5 mL) can be obtained. Considering the operability of the experiment, the optimal conditions of the independent variables are corrected as X_1_ = 10.2, X_2_ = 50.5 °C, X_3_ = 3 h.

This foaming experiment was repeated three times in accordance with what had been predicted by the model, and the average value of the dependent variable was 486 mL, which was very close to what had been predicted by the model.

### 3.3. Performance Evaluation of Additives

Although the maximum foaming performance obtained using the foaming agent prepared under the optimized conditions was 486 mL, the foaming performance of the commercial plant protein foaming agent can reach 700 mL. Therefore, the foaming agent of soybean dreg protein needs to be further optimized. Adding foam stabilizer to foaming agent is a simple and efficient approach to converting the performance of foam [38,39]. FS-50 and FS-3100 are two short-chain fluorocarbon surfactants that have received increasing attention in recent years. Therefore, their performance was omitted in this experiment.

According to the experimental data, adding FS-50 and FS-3100 to foam can improve its viscosity as well as prevent gas penetration [40]. In general, increasing the viscosity of the foam can increase its stability considerably while minimizing its foaming ability because the freshly created foam does not break down as soon as it is formed, as is the case with increasing viscosity. The viscous resistance can, however, be difficult to overcome if one uses a thick solution, which can lead to a decrease in foaming ability [34]. Due to the fact that the foaming agent already had a high foam stability, it was not necessary or obvious to evaluate the effect of the foaming agent on foam stability in this study. In this regard, the surfactant is more appropriate as an additive for protein foaming agents than other substances.

The experimental results can be summarized as follows: the addition of FS-50 and FS-3100 had a good effect on the foaming property of the soybean residue protein foaming agent (Figure 5). Specifically, when the amount of FS-50 was 1.0 g/L, the foaming rate was increased by 172.84%. However, for FS-3100, the highest foaming (810.21 mL) occurred when the addition amount was 1.4 g/L. If the addition amount exceeds these two values, the foaming property of the foam will decrease.

Generally speaking, adding foam stabilizer within as uitable range can reduce the surface tension of the protein solution, so as to improve the foaming property. However, when the foam stabilizer is added in excess, it will have adverse effects. The reason may be that the micelle form of excessive foam stabilizer no longer reduces the surface tension. At the same time, the foam stabilizer can expand the macromolecular chain in protein, and its dispersion effect is remarkable. Foam stabilizer makes protein molecules more evenly dispersed on the surface of foam, which not only improves the rigidity of the foam, but is also favorable for the foaming and stability of the foam [41]. However, when the stabilizer is overdosed, the molecules will remain at the hydrophobic location of the protein, which is unfavorable for the interaction between the reaction groups, thereby reducing the performance of the foam.

To make the right choice between the FS-50 and the FS-3100, it was necessary to carry out a comprehensive analysis (Table 6). FS-50 had a price of 0.258 USD/L, which is cheaper than FS-3100’s price of 0.362 USD/L, and this offered a major advantage in terms of cost when the additives were compared with each other.

Through the evaluation of the cost factors, safety performance, and foam-stabilizing effects of FS-50 and FS-3100, we found that FS-50 is more suitable as a foam stabilizer of the soybean residue protein (SRP) foaming agent. At the same time, the cost of SRP-50 is lower than that of the commercially available plant protein foaming agent (12 USD/L).

### 3.4. Properties of Foams

#### 3.4.1. Density, Viscosity, and Stability of Foams

The foam density, viscosity, and surface tension corresponding to SRP and PS arelisted in Table 7. The foam density of SRP was 7.59% higher than that of PS foam, and the viscosity of the foam increased by 25.66%. Meanwhile, the surface tension of the foam decreased by 9.40%. The above data show that the strength and stability of SRP foam are higher than that of PS [42].

It was shown that approximately 20% of the PS had been drained within the first ten minutes, while roughly 10% of the SRP foam had been drained (Figure 6). In line with previous research [43], it can be deduced that SRP has a higher level of foam stability than PS.

#### 3.4.2. Morphology of Foams

A foam surface could be observed immediately after the OM foam was produced. There are 32.48 μm of foam wall thickness in the diagram and 62.29 μm of foam wall thickness in the actual construction (Figure 7).

### 3.5. Influence of SRP on Foam Concrete

#### 3.5.1. Compressive strength and shrinkage

Figure 8 illustrates the effect of compressive strength on the experimental group (FC-SRP-50) and the control group (FC-PS). During the experiment, both the experimental and control groups showed an increase in compressive strength. Approximately 28 days after the experiment, the compressive strength of the experimental group exceeded that of the control group, which was 2.95 MPa. In conclusion, foam concrete prepared with SRP has a higher compressive strength than foam concrete prepared with PS.

There was a reduction of 3.42 × 10^3^ in dry shrinkage for the control group, as opposed to are duction of 1.75 × 10^3^ for the experimental group on day 90. Therefore, SRP foam concrete is more resistant to shrinkage than PS foam concrete (Figure 9).

#### 3.5.2. Microstructure of the Foam Concrete

Micro-morphologies of the samples were determined using SEM for both the control group and experimental group (Figure 10). There are pores and cracks in the surface of the foam concrete surface in the control group, whereas the surface of the foam concrete in the experimental group is smooth and complete. SRP foam concrete hydrates more effectively than PS foam concrete, as indicated by the experimental results.

#### 3.5.3. Pore Characteristics

Scanners (Beijing Flukes Technology Co., Ltd., Beijing, China) were used to explore the cross sections of the control and experimental groups after 28-day curing. (Figure 11) They were processed using Image Pro Plus software to determine the frequency distribution of pore size and fit the curve based on the sample. Earlier studies [44] have shown that the pore-size distribution of foam concrete is mainly logarithmic in nature, which can explain the results of these studies well [44]. A probability function, f(χ, μ, σ) was used to fit the distribution of pore sizes to the data.
(4)$$f\left(\chi ,\;\mu ,\;\sigma \right)=\frac{1}{\chi \sigma \sqrt{2\pi }}e^{\frac{{\left(\mathrm{lnx}-\mu \right)}^{2}}{2\sigma^{2}}}$$
σ represents the standard deviation and μ denotes the average. χ is the air-void diameter of the sample.

The standard deviations of the control and experimental groups’ fitting curves were 0.39 and 0.25, respectively, while their logarithmic means were 7.84 and 6.12, respectively. It is well established that the pore sizes and distribution of the pore sizes increase with the Landreman’s value [45]. According to the results, the average pore diameter of each sample of the control group was 8.36% of the total diameter of the samples. However, in the experimental group, this proportion increased to 71.5%. The increased viscosity of the SRP solution makes the bubble film stronger. It has two effects: it increases the surface strength of the liquid film, and delays the foam drainage. The improved strength of the film will hinder the diffusion of the internal gas towards the outside and increase the ability of the bubble to resist external disturbances. Additionally, the increased viscosity will increase the resistance forces when the liquid flows in the plateau borders; this can restrict the growth, drainage, and coalescence of the bubbles, and thus, optimize the size distribution of the foam concrete [30]. It is therefore assumed that SRP is able to narrow down the distribution of the pore size of samples, leading to more uniform pore sizes.

### 3.6. Overall Evaluation

In this experiment, soybean dregs were used as raw materials to prepare foaming agents. As far as the literature reports in recent years are concerned, there have been no similar efforts made by scholars. Compared with commercial plant protein foaming agents, SRP-50 produced better foam concrete [7,46]. The reason may be that the effective component of the plant protein foaming agent is triterpenoid saponin, which is a non-ionic surfactant with good air-entraining performance. When it is dissolved in water, the macromolecules are adsorbed on the gas–liquid interface to form a directional arrangement of two groups, which reduces the tension of the gas–liquid interface and makes it easy to produce a new interface. However, previous studies have confirmed that soybean contains a large number of triterpenoid saponins [47]. Meanwhile, soybean residue is a widely distributed biomass material, which has the advantages of being easy to access, having a low cost, and being pollution-free and sustainable. Therefore, in this paper, soybean dregs were successfully used to obtain a high value.

## 4. Conclusions

According to the results of this study, a source of protein foaming agent that can be derived from soybean residue protein was confirmed as a possibility. Bean dregs can be directly made into a soybean residue protein (SRP) foaming agent after heat and alkali treatment. As a result of the preparation of foam concrete and foam using SRP, the following results were obtained:

1. It was determined that the foaming agents are foam able only when the reaction temperature is between 40 °C and 60 °C, the reaction time is between 2 h and 4 h, and the pH value is between 9 and 11, using RSM and BBD. As a result, 50.5 °C was found to be the optimum temperature for the optimum time, and the optimum pH value was 10.2; the highest foam ability (486 mL) was obtained under these conditions.

2. The amount of foaming liquid added to the SRP foaming agent was increased to 343.63 mL by adding 1.0 g/L FS-50, which is a higher amount than that produced by a commercial plant protein foaming agent. Meanwhile, the cost of FS-50 is lower than that of FS-3100 by 1.104 USD/L. Therefore, FS-50 is more attractive.

3. Compared with PS, the compressive strength and shrinkage of foam concrete produced by SRP-50 is improved and the cost is lower.

4. SRP-50 foam concrete was prepared with a narrow pore-size distribution, which may be one of the reasons why it performed better than other foam concretes manufactured using other methods.

In conclusion, the soybean residue protein (SRP) foaming agent produced via the pyrolysis of soybean residue has the advantages of being widely available, low in cost, and high in safety; therefore, it has the potential to be popularized.

## Figures and Tables

**Figure 1 materials-15-07384-f001:**
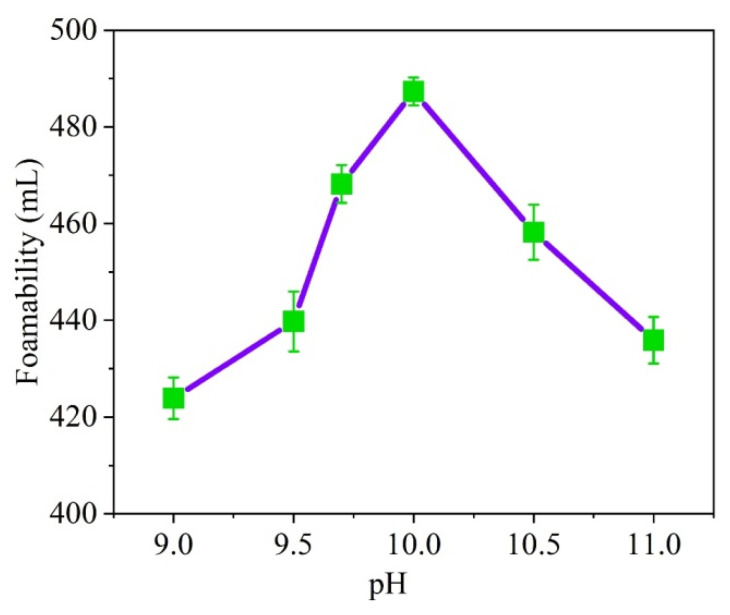
Effect of pH value on foaming ability.

**Figure 2 materials-15-07384-f002:**
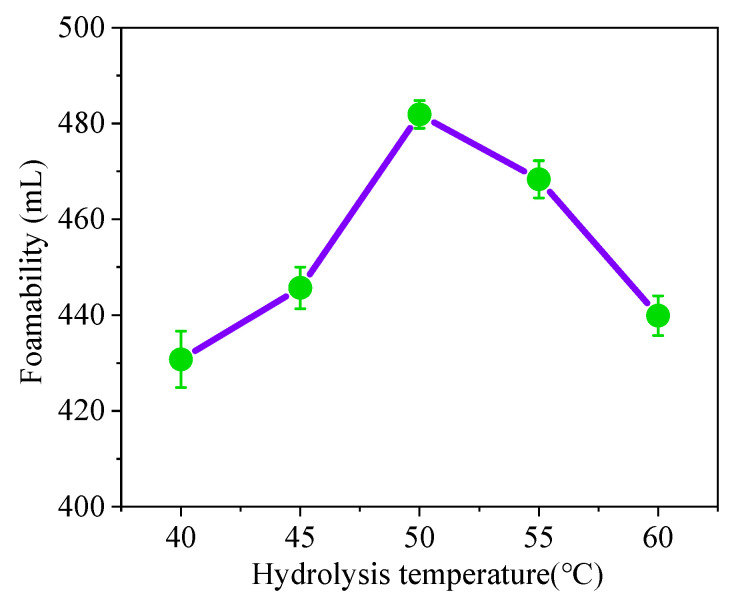
Effect of hydrolysis temperature on foaming ability.

**Figure 3 materials-15-07384-f003:**
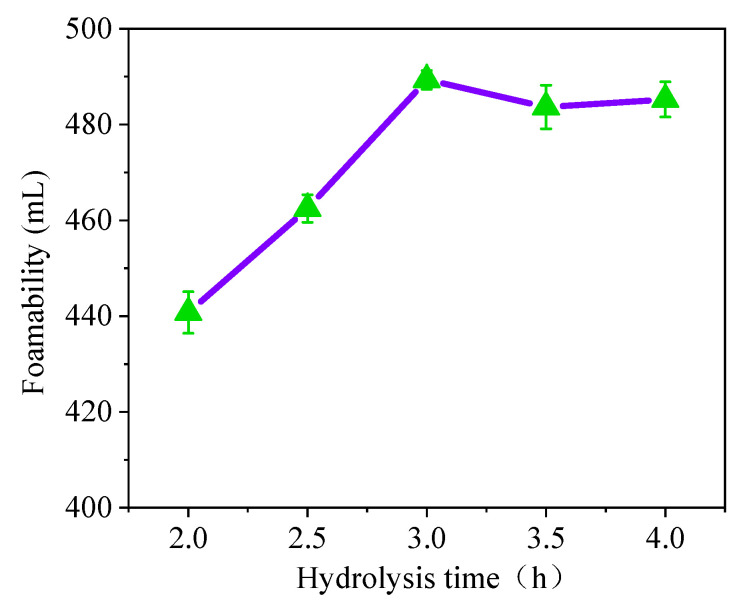
Effect of hydrolysis time on foaming ability.

**Figure 4 materials-15-07384-f004:**
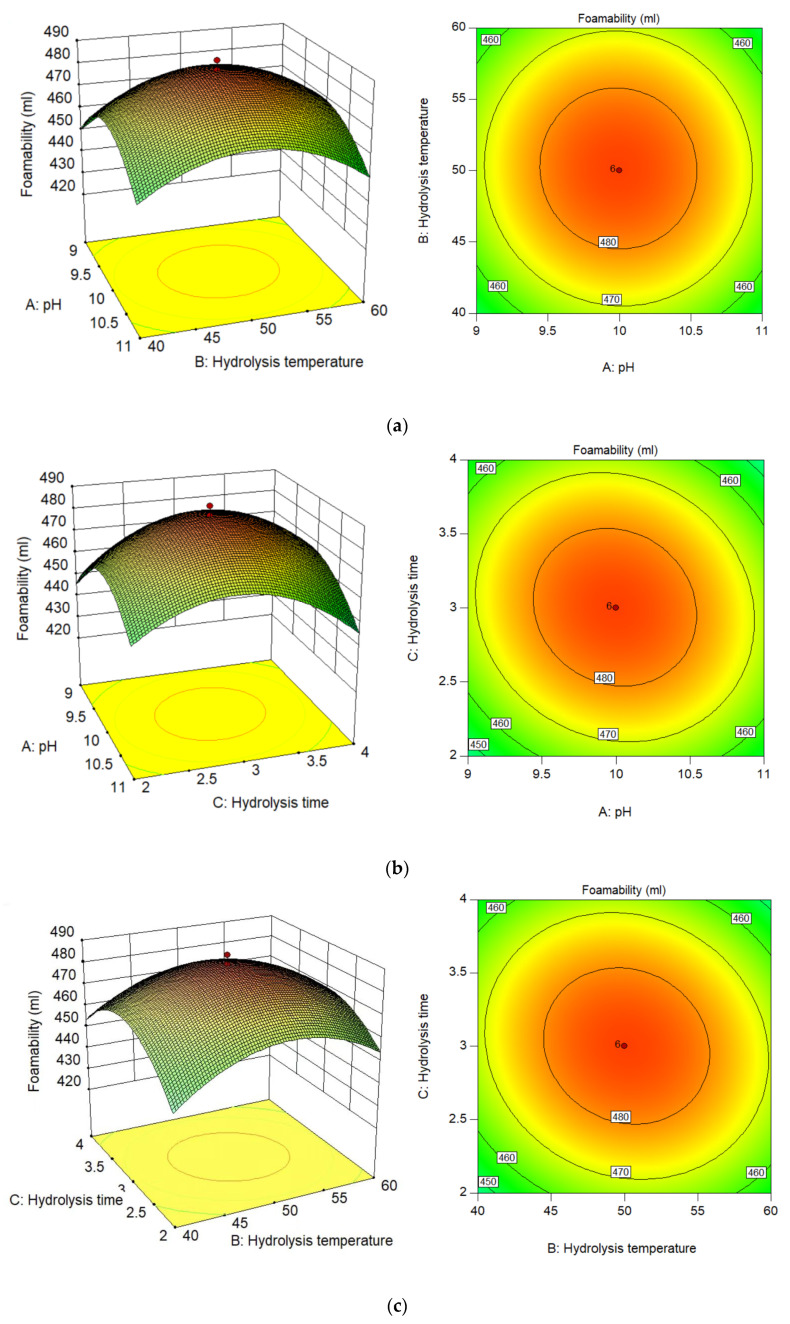
Response surface curves (**left**) and contour plots (**right**): (**a**) fixed X_3_ at 3 h; (**b**) fixed X_2_ at 50 °C; (**c**) fixed X_1_ at 10.

**Figure 5 materials-15-07384-f005:**
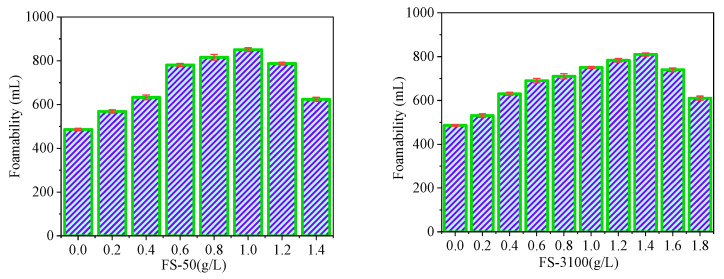
Effects of FS-50 and FS-3100 on foaming ability.

**Figure 6 materials-15-07384-f006:**
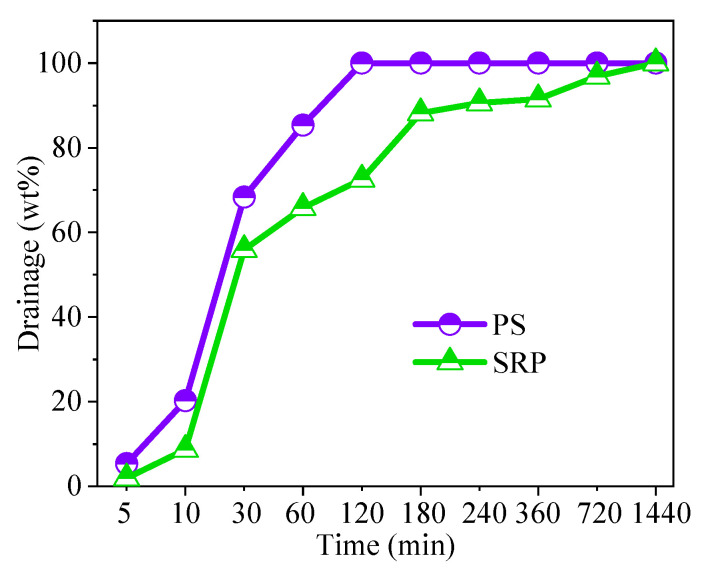
Drainage of foams.

**Figure 7 materials-15-07384-f007:**
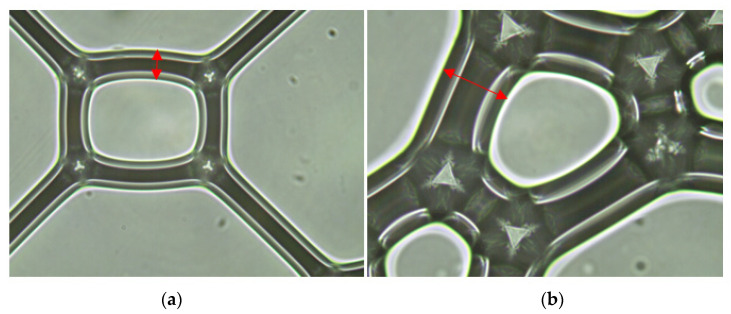
Morphology of foams: (**a**) PS—20 times; (**b**) SRP—20 times.

**Figure 8 materials-15-07384-f008:**
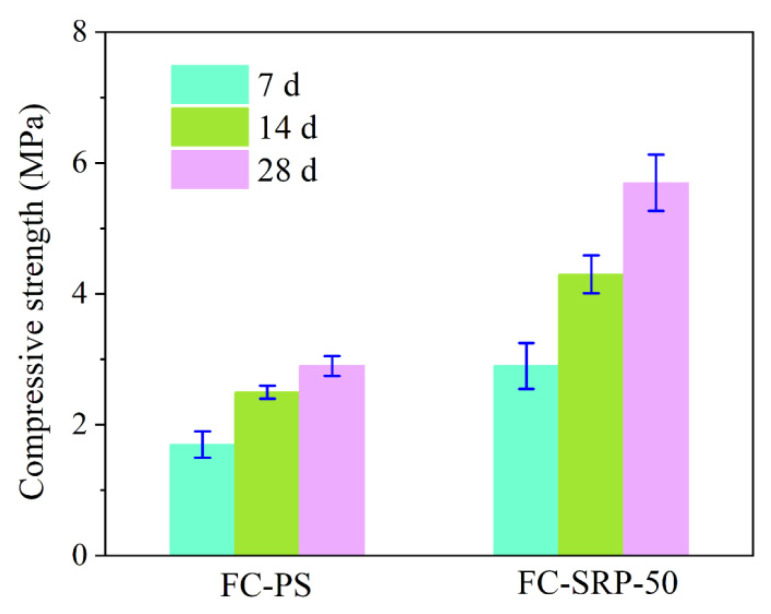
Compressive strength of foam concretes.

**Figure 9 materials-15-07384-f009:**
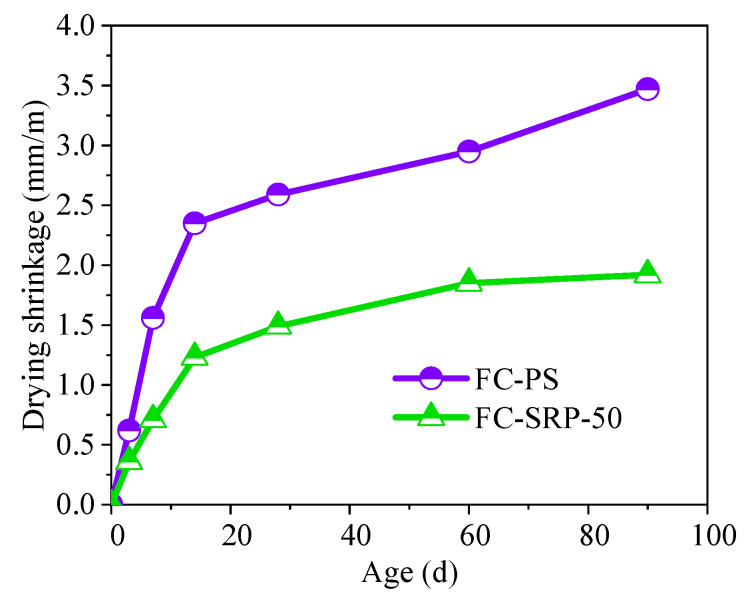
Drying shrinkage of foam concretes.

**Figure 10 materials-15-07384-f010:**
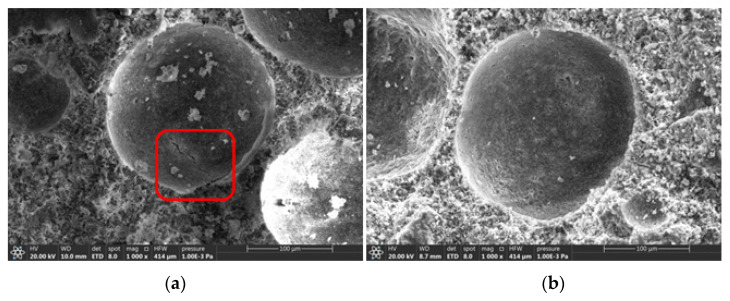
Microstructure of the control group (**a**) and experimental group (**b**).

**Figure 11 materials-15-07384-f011:**
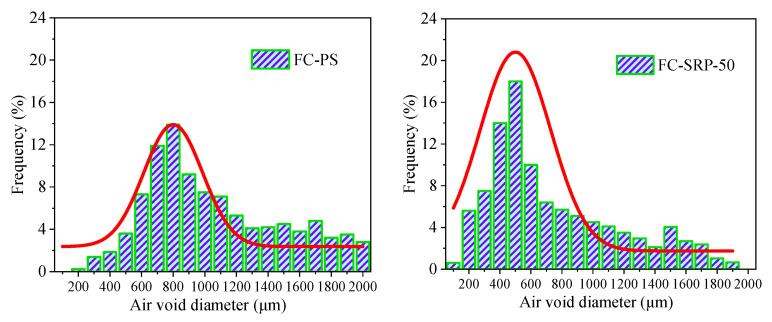
Aperture distributions of foam concrete group.

**Table 1 materials-15-07384-t001:** Compositions of cement, wt.%.

Oxide	CaO	SiO_2_	Al_2_O_3_	Fe_2_O_3_	TiO_2_	MgO	SO_3_	K_2_O	Na_2_O	LOI
Content	64.65	21.32	5.04	3.12	0.97	0.49	2.96	0.39	0.23	3.21

**Table 2 materials-15-07384-t002:** Response surface factors and levels.

Variables	Symbol	Range and Level		
		−1	0	1
pH	X_1_	9	10	11
Hydrolysis temperature (°C)	X_2_	40	50	60
Hydrolysis time (h)	X_3_	2	3	4

**Table 3 materials-15-07384-t003:** Mix design of foam concrete.

Mix	Target Density (kg/m^3^)	Foaming Agent	Cement (kg)	Water (kg)	Foam (m^3^)	Actual Average Dry Density (kg/m^3^)
FC-PS	600	PS	600	300	0.9	607 ± 4
FC-SRP-50	600	SRP	600	300	0.9	613 ± 5

**Table 4 materials-15-07384-t004:** BBD experimental data.

Run	pH	Hydrolysis Temperature (°C)	Hydrolysis Time (h)	Foamability (mL)
1	10	50	3	489.5
2	11	60	4	420
3	10	50	2	430
4	9	40	2	425
5	9	50	3	435
6	10	50	3	480.5
7	10	50	3	485
8	11	40	2	430.5
9	11	40	4	428
10	10	50	3	485.5
11	10	60	3	440
12	9	60	2	435.5
13	10	50	3	483.5
14	10	50	4	438.5
15	11	50	3	440
16	10	40	3	441.5
17	10	50	3	485
18	9	60	4	433.5
19	11	60	2	439
20	9	40	4	435.5

**Table 5 materials-15-07384-t005:** Analysis of variance results.

Source	Sum of Squares	df	Mean Square	F-Value	*p*-Value	Significant
Model	11,245.23	9	1249.47	71.55	<0.0001	**
X_1_ (pH)	0.81	1	0.81	0.05	0.8336	
X_2_ (Hydrolysis temperature)	3.30	1	3.30	0.19	0.6730	
X_3_ (Hydrolysis time)	0.29	1	0.29	0.02	0.8990	
X_1_X_2_	6.46	1	6.46	0.37	0.5566	
X_1_X_3_	103.32	1	103.32	5.92	0.0353	
X_2_X_3_	104.18	1	104.18	5.97	0.0347	
X_1_^2^	4318.46	1	4318.46	247.28	<0.0001	**
X_2_^2^	3887.88	1	3887.88	222.62	<0.0001	**
X_3_^2^	4998.65	1	4998.65	286.23	<0.0001	**
Residual	174.64	10	17.46			
Lack of Fit	136.08	5	27.22	3.53	0.0963	
Pure Error	38.56	5	7.71			
Cor Total	11,419.87	19				
R^2^	0.9847					
R_adj_^2^	0.9709					

Note: ** indicates extremely significant (*p* < 0.01).

**Table 6 materials-15-07384-t006:** Evaluation of FS-50 and FS-3100 as additives for SRP foaming agent.

Additive	Price (USD/kg)	Dosage (kg/L)	Cost (USD/L)	Economy	Safety	Foaming Ability
FS-50	258	1.0 × 10^−3^	0.258	***	**	***
FS-3100	258	1.4 × 10^−3^	0.362	*	**	**

Note: * stands for inferior; ** stands for medium, and *** stands for superior.

**Table 7 materials-15-07384-t007:** Properties of the foams.

Type	Density (kg/m^3^)	Viscosity (Pa∙S)	Surface Tension (mN/m)
PS	17	0.113	35.1
SRP	22	0.142	31.8

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
