# Peer review of "Preparation and Application of Foaming Agent Based on the Compound System of Short-Chain Fluorocarbon and Soybean Residue Protein"

_materials, 2022, doi:10.3390/ma15207384_

Round 1

Reviewer 1 Report

I review the manuscript and I listed my comments in the pdf file. Although the manuscript is already in the template and its type is correct, it is still full of typos, most of them were marked by me. The manuscript can be accepted after major revision and I suggest extensive editing of the English.

Author Response

Thank you for your valuable comment and suggestion. I feel very sorry and confused. I don't know what causes a lot of similar errors in the uploaded version. I have revised the whole paper, and I'm sorry again.

Reviewer 2 Report

The manuscript entitled "Preparation and application of foaming agent based on the compound system of a short-chain fluorocarbon and soybean residue protein" investigated the design and properties of a foaming agent including soybean residue protein and its applicability as a protein source. Different analyses were applied to characterize the foaming agent as well as modeling with response surface methodology in order to obtain better results. Although the organization of the paper is good, there are some critical points that took my attention and need revision before evaluation for publication in this prestigious journal. Please find below my comments and suggestions;

- The formatting of the manuscript should be revisited following the rules of the MDPI journal group.

-Serious errors are present in the usage of English throughout the whole manuscript like Line 92.. "cumbersomeextraction" should be edited as "cumbersome extraction". And many similar mistakes should be corrected.

-Line 58, can you please refer to the statement "..by reducing the surface tensions of foaming liquid.

- Line 69.. environmentally friendly..

-Line 82.. produced.

-Line 88... decomposed.

-Although some useful information was given in the introduction section, in my opinion, an additional statement for showing the novelty of this study will make this paper better.

- In section 2.2.2, it was mentioned that prior to the single factor test, the effects of different parameters such as pH, temperature, and hydrolysis time were investigated. I think it will be better to add a statement if these experiments were carried out in the order of pH, temperature, and time. Because in its current form, it is not clear how the other parameters were fixed while focusing on a new parameter.

In section 2.3, if possible, can you please refer to the procedure followed during experimental studies?

Section 2.4.2. Line 197, is that Du Nouy ring method?

Section 2.4.4. can you please write which microscope (and its properties) was used for optical measurements?

In section 3.1.1., although 4 points of pH were examined for the effect of pH, it was shown and mentioned in the text that there is a critical increment upon increasing the pH value from 9.5 to 10.0. If possible, I think it will be very good to make an additional test at an average value between these points like 9.7 in order to prove that critical increment up to pH 10.0.

In section 3.5.2, can you please explain Figures 10a and b in the text? because as it can be seen in the Figures, the control group seems to be rougher than the experimental one which can be effective in different processes.

Author Response

Dear Editors and Reviewers:

Thank you for your letter and for the reviewers’ comments concerning our manuscript entitled “Preparation and application of foaming agent based on the compound system of short chain fluorocarbon and soybean residue protein” (ID: materials-1942025). These comments, suggestions are very helpful for revising and improving our paper, as well as the important guiding significance to our researches. We have studied comments carefully and have made correction which we hope meet with approval. Revised portion aremarked in red in the manuscript. The main corrections in the paper and the responds to the reviewer’s comments are as following:

Responds to the reviewer’s comments:

Reviewer #2:

  1. - The formatting of the manuscript should be revisited following the rules of the MDPI journal group.

-Serious errors are present in the usage of English throughout the whole manuscript like Line 92.. "cumbersomeextraction" should be edited as "cumbersome extraction". And many similar mistakes should be corrected.

-Line 58, can you please refer to the statement "..by reducing the surface tensions of foaming liquid.

- Line 69.. environmentally friendly..

-Line 82.. produced.

-Line 88... decomposed.

Response:

Thank you for your valuable comment and suggestion. I feel very sorry and confused. I don't know what causes a lot of similar errors in the uploaded version. I have revised the whole paper, and I'm sorry again.

  1. Although some useful information was given in the introduction section, in my opinion, an additional statement for showing the novelty of this study will make this paper better.

Response:

Thank you for your valuable comment and suggestion. The modifications have been done in the revised manuscript according to your comments as below:

Page 4-5, line 90-100:

“This study proposed a new method to obtain protein source only by thermal alkali treatment of soybean residue, without laborious and cumbersome extraction or purification process [23-25]. At the same time, it has the advantages of reusability and environmental sustainability. In order to optimize reaction conditions such as temperature, pH, and time needed for thermal alkali treatment, we adopted the response surface methodology (RSM) to meet the desired purpose. An additive suitable for soybean residue protein (SRP) was found in this experiment to enhance the foaming performance of soybean residue protein (SRP) foaming agent. However, previous research only stayed at the level of preparing protein foaming agents. Thus, this study has investigated not only the foam but also the foamed concrete to further optimize the products of SRP foaming agents.”

  1. In section 2.2.2, it was mentioned that prior to the single factor test, the effects of different parameters such as pH, temperature, and hydrolysis time were investigated. I think it will be better to add a statement if these experiments were carried out in the order of pH, temperature, and time. Because in its current form, it is not clear how the other parameters were fixed while focusing on a new parameter.

Response:

Thank you for your valuable comment and suggestion. The modifications have been done in the revised manuscript according to your comments as below:

Page 6, line 127-132:

“2.2.2. Single factor test

 For the foaming ability, a single factor test was tested to check effect of pH value (9, 10, 11) temperature value (40, 50, 60 ℃), and time of hydrolysis (2, 3, 4 h) were considered. During the single factor test, we checked in order of pH value, temperature value and hydrolysis time. That is to say, when the temperature value and hydrolysis time were concerned, the pH value has been fixed.”

  1. In section 2.3, if possible, can you please refer to the procedure followed during experimental studies?

Response:

Thank you for your question. The modifications have been done in the revised manuscript according to your comments as below:

Page 9-10, line 191-192:

“The procedure followed in the experimental study is Chinese standard “Foam Concrete” (JG/T 266–2011).”

  1. Section 2.4.2. Line 197, is that Du Nouy ring method?

Response:

Thank you for your question. Yes, Du Nouy ring method was used to test the surface tension of foaming agent in this test.

  1. Section 2.4.4. can you please write which microscope (and its properties) was used for optical measurements?

Response:

Thank you for your question. The modifications have been done in the revised manuscript according to your comments as below:

Page 10, line 206-208:

“2.4.4. Optical microscopy

The thickness of the foam wall (OM) was determined using an optical microscope ( DM 750 ) in order to determine the thickness of the wall.”

  1. In section 3.1.1., although 4 points of pH were examined for the effect of pH, it was shown and mentioned in the text that there is a critical increment upon increasing the pH value from 9.5 to 10.0. If possible, I think it will be very good to make an additional test at an average value between these points like 9.7 in order to prove that critical increment up to pH 10.0.

Response:

Thank you for your valuable comment and suggestion. According to your suggestion, we conducted a group of experiments to confirm that critical increment up to pH 10.0. The modifications have been done in the revised manuscript according to your comments as below:

Page 12, line 235:

Fig.1. Effect of pH value on foaming ability

  1. In section 3.5.2, can you please explain Figures 10a and b in the text? because as it can be seen in the Figures, the control group seems to be rougher than the experimental one which can be effective in different processes.

Response:

Thank you for your question. It can be seen from the figure that the control group (a) should indeed be rougher than the experimental group (b). The most important thing is that there are cracks in the samples of the control group, which is the reason for the low strength of the control group. However, in the experimental group, the foamed concrete prepared with self-made foaming agent (SRP-50) is complete, so its strength is high.

  • (b)

Fig. 10. Microstructure of the control group and experimental group

******************************************

We tried our best to improve the manuscript and made some corrections in the paper. In addition to the above revisions, we made some changes in the revised manuscript to improve its quality as well. These changes are not listed in this document but are marked in red in the revised paper. We appreciate for Editors’ and Reviewers’ warm work earnestly, and hope that this revised version will be approved. Once again, thank you very much for your positive comments and constructive suggestions.

Round 2

Reviewer 1 Report

My first review contained few questions (in Pdf comments), but the authors skipped these and I did not get the answers. My questions remained the same. Grammatical errors in the manuscript have been corrected.

Reviewer 2 Report

Following the revisions of reviewevers, I think the paper became better and can be acceptable in its current form.

Author Response

Thank you.

Reviewer 3 Report

Accept

Author Response

Thank you.